# Cutaneous Delivery and Biodistribution of Cannabidiol in Human Skin after Topical Application of Colloidal Formulations

**DOI:** 10.3390/pharmaceutics16020202

**Published:** 2024-01-30

**Authors:** Maria Lapteva, Jonathan Faro Barros, Yogeshvar N. Kalia

**Affiliations:** 1School of Pharmaceutical Sciences, University of Geneva, CMU-1 rue Michel Servet, 1211 Geneva, Switzerlandjonathan.faro@unige.ch (J.F.B.); 2Institute of Pharmaceutical Sciences of Western Switzerland, University of Geneva, CMU-1 rue Michel Servet, 1211 Geneva, Switzerland

**Keywords:** cannabidiol, topical formulation, skin delivery, cutaneous biodistribution profile

## Abstract

The objective of this study was to investigate the cutaneous delivery of cannabidiol (CBD) from aqueous formulations developed for the targeted local treatment of dermatological conditions. CBD was formulated using a proprietary colloidal drug delivery system (VESIsorb^®^) into an aqueous colloidal solution at 2% (ACS 2%) and two colloidal gels (CG 1% and CG 2%, which contained 1% and 2% CBD, respectively). Two basic formulations containing CBD (5% in propylene glycol (PG 5%) and a 6.6% oil solution (OS 6.6%)) and two marketed CBD products (RP1 and RP2, containing 1% CBD) were used as comparators. Cutaneous delivery and cutaneous biodistribution experiments were performed using human abdominal skin (500–700 µm) under infinite- and finite-dose conditions with 0.5% Tween 80 in the PBS receiver phase. The quantification of CBD in the skin samples was performed using a validated UHPLC-MS/MS method and an internal standard (CBD-d3). The cutaneous deposition of CBD under finite-dose conditions demonstrated the superiority of CG 1%, CG 2%, and ACS 2% over the marketed products; CG 1% had the highest delivery efficiency (5.25%). Cutaneous biodistribution studies showed the superiority of the colloidal systems in delivering CBD to the viable epidermis, and the upper and lower papillary dermis, which are the target sites for the treatment of several dermatological conditions.

## 1. Introduction

Preclinical studies point to the potential therapeutic applications of cannabinoids for the treatment of dermatological conditions [1,2,3,4]. However, clinical data on the mechanism of action and the therapeutic efficacy (and hence, clinical relevance) are scarce. The potentially beneficial effects would most likely be mediated via the cannabinoid receptors, CBR1 and CBR2, which are both found in the skin [4]. Alternative targets with which cannabinoids can interact are Transient Receptor Potential (TRP) ion channels and Peroxisome Proliferator-Activated Receptors (PPARs) [4,5]. Through the interaction with these receptors, cannabinoids, in general, and cannabidiol (CBD), in particular, hold promise for the treatment of disorders, such as acne vulgaris [6,7,8,9,10], allergic contact dermatitis [7,10,11], asteatotic dermatitis [12], atopic dermatitis [4,13,14,15,16], hidradenitis suppurativa [17], Kaposi sarcoma [10,18,19], pain [20,21,22,23,24], pruritus [7,12,25,26,27,28,29,30,31,32], psoriasis [7,33], skin pigmentation disorders [34], and skin cancer [7,35], and can serve to improve skin protection, barrier function [36,37,38,39,40], and wound healing outcomes [41,42,43]. CBD is thought to act primarily as a modulator of inflammatory processes in the skin [44] and can reduce itching [45]. It has also been reported that cannabinoids, including CBD, show promise for the treatment of skin infections due to their anti-microbial properties [46,47]. Further information about the various pathways involved is provided in a review by Tóth et al. [5].

As mentioned above, despite promising preclinical data, reliable human in vivo data on the benefits of CBD are lacking. This is due in part to the fact that CBD’s physicochemical properties render it a formulation challenge for topical or transdermal drug delivery. Although CBD has a molecular weight of 314.47 Da and a melting point of 66–67 °C, which are both compatible for delivery via the skin, it is highly lipophilic (log P 5.79), has very poor aqueous solubility, and is highly sensitive to degradation when exposed to light, heat, oxygen, or an alkaline pH [1,2]. A possible strategy to facilitate the topical or transdermal delivery of CBD involves its incorporation into more advanced drug delivery systems able to mask or overcome its drawbacks [48,49].

To this end, colloidal drug delivery systems (CDDS) may represent an attractive option for the topical delivery of CBD [50,51,52,53]. CDDS have already been shown to significantly improve the oral drug delivery of a number of drugs and nutraceuticals, improving their bioavailability several fold [54,55,56,57,58,59,60,61,62]. These improvements in bioavailability are dependent on small droplet sizes and a uniform droplet size distribution [63,64,65,66,67,68,69,70,71,72,73] because large, inhomogeneous colloidal formulations can lead to an unpredictable and poor reproducibility outcome [55,74]. However, enhanced bioavailability alone cannot be considered a real improvement when a drug delivery system fails to deliver clinically relevant concentrations of a drug [75,76] to the therapeutic target or when appreciable amounts of the drug are delivered, but no beneficial clinical effects are observed [77].

Due to its very small and highly reproducible mean droplet sizes and the very low polydispersity index, VESIsorb^®^ Formulation Technology-based CDDS (VESIsorb CDDS) have been highly successful in improving the oral bioavailability of a number of poorly water soluble compounds, such as coenzyme Q_10_ (CoQ10) [75], CBD [78], and beta-caryophyllene (BCP) [79]. Moreover, the efficacy and safety of the technology for topical delivery to the skin have been demonstrated by its longstanding use for the cutaneous delivery of Natural Moisturizing Factor (NMF) and D-panthenol [80,81]. The safety, tolerability, and biocompatibility of the delivery system were further demonstrated in a proof-of-concept study to treat dry eye disease by the topical ocular delivery of a VESIsorb CDDS containing ectoin [82].

The objective of the present study was to use this technology to develop an aqueous colloidal solution of CBD and to measure CBD delivery into and across human skin with regard to dermatological applications. Hence, the cutaneous deposition of CBD was a far more relevant parameter than transdermal permeation for formulation evaluation, and indeedm, selectivity for deposition was highly desirable. Although infinite-dose conditions were used for the initial studies, finite dosing was used in the second part of the study to mimic more closely the actual conditions of use in vivo. For the finite-dose studies, CBD gel formulations—at 1% and 2%—were also prepared since this was clearly a more patient-friendly and practical dosage form. In order to benchmark CBD delivery achieved using these formulations, it was decided to use two commercial/marketed products as references. In addition to the quantification of the cutaneous deposition and transdermal permeation of CBD, its cutaneous biodistribution was also determined—but only under finite-dose conditions—in order to determine the amounts of CBD present as a function of penetration depth in the skin. This provided an insight into the CBD concentrations that could be achieved in the different anatomical regions in the skin and would obviously be of interest with respect to the feasibility of using topical administration for the targeted local treatment of dermatological conditions.

## 2. Materials and Methods

### 2.1. Materials

Cannabidiol (CBD) used in the study was extracted from the aerial parts of *Cannabis sativa* L. (hemp) and was concentrated to a CBD content of 90% with a tetrahydrocannabinol (THC) content below 0.3%. Thus, this extract was considered as “full-spectrum hemp oil” in the field of cannabis science. Deuterated CBD (CBD-d3) was used as an internal standard and was obtained from Sigma Aldrich (Merck KGaA, Darmstadt, Germany). Other reference compounds, such as THC and cannabigerol (CBG), were procured from Lipomed AG, (Arlesheim, Switzerland) and Supelco (Merck KGaA, Darmstadt, Germany). Acetonitrile (ACN) and methanol (MeOH) (HPLC grade and LC/MS grade) were received from Fisher Scientific (Reinach, Switzerland). Isopentane, Dulbecco’s phosphate-buffered saline (without calcium chloride or magnesium chloride; DPBS); Tween 20 and Tween 80 were purchased from Sigma Aldrich (Merck KGaA, Darmstadt, Germany). Formic acid (extra pure 99%) (FA) was obtained from Biosolve Chemicals (Dieuze, France). Brij™ C20-PA-(RB) was purchased from Croda Europe (East Yorkshire, UK). O.C.T. embedding matrix was sourced from CellPath (Newtown, UK). Ultrapure water (Millipore Milli-Q Gard 1 Purification Pack resistivity > 18 MΩ·cm; Zug, Switzerland) with a filter (Millipak^®^ 40 Millipore) of 0.22 μm was used to prepare all the solutions. The marketed CBD-containing products, Charlotte’s Web Cream 1% (RP1 1%) and Lord Jones Serum (Face-oil) 1% (RP2 1%), were supplied by Vesifact AG (Baar, Switzerland).

### 2.2. Analytical Methods

#### 2.2.1. Quantification by UHPLC-UV

UHPLC coupled with UV detection was used to determine the CBD content of the different formulations during stability testing. The UHPLC method was a modification of the low-pressure isocratic HPLC method described by Sigma-Aldrich (https://www.sigmaaldrich.com/CH/de/technical-documents/protocol/analytical-chemistry/small-molecule-hplc/analysis-of-cannabinoids, accessed on 10 January 2024) using a 2.1 × 50 mm Acquity UPLC^®^ BEH C18 1.7 µm column. The mobile phase comprised 0.1% formic acid in ultrapure water and ACN (20:80). CBD was detected using an Acquity UPLC^®^ PDA eLambda detector at a wavelength of 220 nm. The limit of detection and the limit of quantification were 0.1 µg/mL and 0.5 µg/mL, respectively. The retention time for CBD was 1.15 min. The CBD Aqueous Colloidal Solution (ACS) and reference formulations were diluted to a nominal CBD concentration of 15 µg/mL, and were subsequently measured in order to define the total drug content in the formulation and the deviation of the actual drug content from the nominal value.

#### 2.2.2. Quantification by UHPLC-MS/MS

UHPLC with tandem mass spectrometry detection (UHPLC-MS/MS) was used to quantify CBD delivery to and across the skin during the cutaneous delivery studies. A Waters Acquity UPLC^®^ system (Baden-Dättwil, Switzerland) equipped with a binary solvent pump and sample manager coupled to a Waters XEVO^®^ TQ-S micro MS tandem quadrupole detector (Baden-Dättwil, Switzerland) were used.

Gradient separation was carried out using a Waters XBridge BEH C18, 2.1 × 50 mm column containing 2.5 μm particles. The gradient conditions are described in Table 1. Mobile phase A consisted of ultrapure water and mobile phase B consisted of a mixture of MeOH:ACN (50:50 with 0.1% formic acid). The flow rate was set at 0.3 mL/min and the column was maintained at 35 °C. All samples were injected using an injection volume of 2 µL. CBD and CBD-d3 were co-eluted at a retention time of 2.27 min (complete details are presented in the Appendix A).

Mass spectroscopy detection was performed with electrospray ionization using the Multiple-Reaction Monitoring (MRM) mode. To account for the matrix effect, each injected sample contained an internal standard (CBD-d3) at approximately a 40 ng/g concentration. The different UHPLC-MS/MS settings for the analytes are presented in Table 2.

MassLynx software (V4.1, Waters Inc., Milford, MA, USA) was used for data integration and analysis. The validation of the analytical method was performed as per current EMA [83] and ICH guidelines [84]: the specificity, sensitivity, linearity, accuracy, and precision were tested with the goal of quantifying CBD in skin extracts and permeation samples. The complete details are presented in the Appendix A.

### 2.3. Development of Formulations

#### 2.3.1. Development of the Aqueous Colloidal Solution (ACS) Formulation

A novel colloidal formulation based on the proprietary VESIsorb CDDS was developed for the topical delivery of CBD, ACS 2% (Table 3). These colloids are droplet-like structures with a lipid/triglyceride core that is the primary carrier for the lipophilic CBD. This core is surrounded by a monolayer of amphiphilic molecules, such as phospholipids and, optionally, co-surfactants. The unique characteristics of colloids are their small droplet size and homogenous droplet size distribution. The ACS 2% formulation was characterized by measuring the size of the droplets using a Zetasizer Nano S instrument (Malvern Instruments Limited, Worcestershire, UK).

#### 2.3.2. Development of the Colloidal Gel Formulation

In order to identify the best gelling agent, the ACS 2% solution was gelled by adding a series of thickening agents (carbomer, carrageenan, hydroxypropylmethylcellulose (HPMC), and xanthan gum) to the ACS 2% at increasing concentrations (e.g., 1.0%, 1.5%, and 2.0% *w*/*w*) under magnetic stirring. Based on the results of these preliminary studies, xanthan gum was selected for the preparation of the gel formulations used in the finite-dose studies—CG 1% and CG 2% (Table 4) containing 1% and 2% CBD, respectively. For the preparation of CG 1%, the ACS 2% solution was first diluted at a 1:1 (*w*/*w*) ratio with the buffer used to prepare ACS 2% and was subsequently gelled.

#### 2.3.3. Development of the Comparator Formulations

Two basic formulations were prepared with the aim to enable a comparison of the delivery of the CBD in this study with the data in the published literature. Thus, two simple solutions of either 5% CBD in propylene glycol or 6.6% CBD in a mixture of medium- and long-chain triglycerides and propylheptyl caprylate were prepared by dissolving the CBD in the respective solvent or solvent mixture. The resulting basic formulations were characterized by visual appearance and CBD content.

### 2.4. Human Skin Preparation

Human skin was used for the skin deposition/permeation experiments. Samples were collected immediately after surgery from the Department of Plastic, Aesthetic and Reconstructive Surgery, Geneva University Hospital (Geneva, Switzerland). The study was approved by the “Commission cantonale d’éthique de la recherche” (CCER—2021-01578). The hypodermis and fatty tissue were removed and the skin was sliced to a uniform thickness (500–700 µm) using a Thomas Stadie-Riggs manual skin slicer (Thomas Scientific, Swedesboro, NJ, USA) and stored at −20 °C until use or for a maximum period of 3 months.

### 2.5. Preliminary Studies

To ensure appropriate experimental conditions, preliminary studies were performed in order to determine (i) the optimum composition of the receptor phase in the Franz cell receiver compartment to ensure sufficient CBD solubilization so that sink conditions were maintained, (ii) the stability, and (iii) extraction efficiency of the skin samples. All the methods and detailed results are presented in the Appendix A.

### 2.6. Evaluation of Skin Delivery In Vitro

#### 2.6.1. CBD Delivery under Infinite-Dose Conditions

Table 3 summarizes the different formulations tested and the experimental conditions for the infinite-dose experiments.

Before starting the experiment, the skin samples were thawed at room temperature and equilibrated in a 0.9% NaCl solution for 15 min. Skin integrity was monitored using a Vapometer (Delfin Technologies Ltd., Kuopio, Finland) at the beginning of the experiment.

Human skin samples were mounted in 2.0 cm^2^ Franz cells, which were then filled with the selected receiver solution (10 mL of 0.5% Tween 80 in PBS). Formulations (500 mg/cm^2^; infinite dose) were applied to each Franz cell and the donor compartment was covered with parafilm to ensure occlusion. The receiver compartments were stirred at 250 rpm and maintained at 32 °C using a water bath. Each condition was tested with 6 replicates.

Aliquots (0.3 mL) were withdrawn from the receiver compartment at pre-determined time points: 10, 12, 18, 24, 36, 42, and 48 h. Each aliquot was replaced with an equal volume of fresh buffer. Permeation samples were centrifuged at 10,000 rpm for 20 min prior to CBD quantification after the appropriate dilution and addition of the internal standard.

At the end of the experiment, the formulations were removed from the skin surface with tissue paper, and the skin surface was cleaned with a cotton swab moistened with 0.5% Tween 80 in PBS and rubbed twice to remove any residue. Then, the skin area in contact with the formulation (~2.0 cm^2^ disc) was punched out from the skin samples. These samples were cut into small 1 mm × 1 mm pieces and subjected to CBD extraction (10 mL of MeOH:H_2_O 90:10) for 4 h. The extraction samples were centrifuged at 10,000 rpm for 20 min before CBD quantification after the appropriate dilution and addition of the internal standard. All additions of volume and internal standards to the sample as well as the dilution were performed by weight measurement. The drug contents of the different formulations were expressed according to the nominal value.

#### 2.6.2. CBD Delivery under Finite-Dose Conditions and Investigation of CBD Biodistribution

Table 4 summarizes the different formulations tested and the experimental conditions used for the finite-dose experiments.

Human skin samples were prepared as previously described and mounted in 2.0 cm^2^ Franz cells, which were then filled with the selected receiver solution (10 mL of Tween 80 0.5% in PBS). Formulations (15 mg/cm^2^; finite dose) were applied to each Franz cell and the donor compartments were kept unoccluded. The receiver compartments were maintained under constant stirring at 32 °C using a water bath. Each condition was tested with 6 replicates. Aliquots (0.3 mL) were withdrawn from the receiver compartment at pre-determined time points: 10, 16, 20, and 24 h. Each aliquot was replaced with an equal volume of fresh buffer. Permeation samples were centrifuged at 10,000 rpm for 20 min before CBD quantification after the appropriate dilution and addition of the internal standard.

At the end of the experiment, a small area of 0.5 cm^2^ was punched out from the 2 cm^2^ skin samples. These skin discs were snap-frozen in isopentane cooled with dry ice (−78.5 °C) followed by cryotoming (Thermo Scientific CryoStarTM NX70; Reinach, Switzerland) to obtain 10 lamellae with a thickness of 40 µm encompassing the stratum corneum, viable epidermis, and upper and lower dermises down to a depth of ~400 µm. The CBD deposited in each lamella was extracted in 200 µL of MeOH:H_2_O 90:10 overnight with continuous stirring at room temperature. The extraction samples were centrifuged at 10,000 rpm for 10 min and diluted prior to UHPLC-MS/MS analysis. A small area of 0.28 cm^2^ was also punched out of the remaining skin, trimmed into small 1 mm × 1 mm pieces, and subjected to CBD extraction and subsequent quantification to determine the total skin deposition. All additions of volume and internal standards to the sample as well as dilutions were performed by weight measurement. The drug content of the different formulations and drug delivery efficiencies calculated were expressed according to the nominal value.

### 2.7. Data Analysis

The data are expressed as the mean ± SD. The data were processed and evaluated statistically using GraphPad Prism 9.4.1. Groups were compared using either an analysis of variance (ANOVA) or analysis of means by the Student’s *t*-test. Tukey’s multiple comparisons test or Bonferroni *t*-test were used, when necessary, as post hoc procedures. The level of significance was fixed at α = 0.05.

## 3. Results and Discussion

### 3.1. Analytical Methods

The analytical methods for the quantification of CBD were shown to be specific, sensitive, linear, accurate, and precise. Thus, in accordance with the current EMA [83] and ICH guidelines [84], they were considered as valid for the quantification of the CBD in the range of 2.3 to 100.0 ng/g for the skin extraction samples and in the range of 1.2 to 100.0 ng/g for the skin permeation samples. The complete results and details are presented in the Appendix A.

### 3.2. Formulation Development and Characterization

#### 3.2.1. Aqueous Colloidal Solution Formulation

The ACS 2% formulation was prepared and the total drug content in the formulation was measured to be in the range of 20.82 to 21.03 mg/g, corresponding to a deviation of between +4.0 and +5.2% as compared with the expected nominal values. The total CBD loading efficiency was 99.63% and ACS 2% presented an excellent two-year stability at 25 °C as assessed by the visual appearance, colloidal droplet size, and CBD content (neither creaming nor the sedimentation of CBD crystals were observed, despite the extremely low aqueous solubility of CBD (~0.1 μg/mL [85]) (Table 5)). The mean diameter of the droplets formed was between 45 to 50 nm and the size distribution of the droplets was homogeneous, exhibiting a single main population and a polydispersity index of <0.100. Importantly, these colloids could be sterile-filtered and thus produced without the need for preservatives.

#### 3.2.2. Colloidal Gel Formulation

The colloidal gel formulations (CG 1% and CG 2%) were prepared as described above, and the CBD content was determined by UHPLC-UV. The experimentally determined CBD contents for the nominal 1% and 2% CBD colloidal gels were 10.51 and 20.64 mg/g, corresponding to a deviation from the nominal values of +5.1% and +3.2%, respectively. The CBD loading corresponded to efficiencies of 99.62% and 97.83% for CG 1% and CG2%, respectively. The storage stability of CG 2% was also at least 24 months at 25 °C, as assessed by the visual appearance, colloidal droplet size, and CBD content (Table 5). The gelation did not impair CBD stability in the formulation and had no influence on the colloidal droplet size (Table 5). However, the gel formulation was characterized by a slightly larger PDI compared to the aqueous colloidal solution, but this increase in the PDI was expected due to the semi-flexible nature of xanthan gum and its tendency to self-associate [86].

#### 3.2.3. Basic Comparator Formulations

The CBD content in the two basic comparator formulations (PG 5% and OS 6.6%) was measured by UHPLC-UV as well. For PG 5%, the drug content ranged from 50.18 to 50.81 mg/g, which corresponded to a deviation from the nominal concentration by +0.4% to +1.6%. For OS 6.6%, the measured CBD content ranged from 66.52 mg/g to 66.66 mg/g, which amounted to 99.88% to 100.00% of the nominal value.

#### 3.2.4. Reference Formulations

The CBD content of the two reference products (RP1 1% and RP2 1%) was measured by UHPLC-UV. For RP1 1% (Charlotte’s Web Cream 1%), the drug content ranged from 12.40 to 14.27 mg/g, deviating from the nominal concentration by +24.0% to +42.7%. For RP2 1% (Lord Jones Serum face oil 1%), the measured CBD content was 12.20 mg/g, amounting to +22.0% above the nominal value.

### 3.3. Preliminary Studies

The optimal receiver medium was determined to be PBS with 0.5% Tween 80. Detailed results can be found in the Appendix A. The extraction method was considered as being efficient since 98.8 ± 22.9% of the CBD was recovered upon skin extraction (mean ± SD, *n* = 8). After contact with skin, CBD was quantified by UHPLC-MS/MS. No known degradation compounds were detected. The relative amount of CBD recovered as a function of time is shown in the Appendix A. At the end of the experiment, 99.5 ± 17.7% of CBD was recovered. Based on the results, it was concluded that CBD was stable when in contact with the skin at 32 °C for at least 48 h, i.e., sufficient to proceed with skin delivery experiments.

In vitro releases studies were not performed since the results were not predictive of the release from a formulation into the stratum corneum, which is a highly hydrophobic environment as opposed to the hydrophilic release medium. Furthermore, for SEDDS, most in vitro methods, i.e., membrane diffusion methods (such as uses of filters, dialysis membranes, or separation methods (e.g., ultracentrifugation)), influence release kinetics [87].

### 3.4. CBD Delivery under the Infinite-Dose Condition

#### 3.4.1. Experimental Observations

All skin samples were considered acceptable with a TEWL below 15.0 g/m^2^h. The measured skin thickness values of all the samples ranged between 500 and 700 μm

#### 3.4.2. CBD Skin Deposition

First, the CBD skin deposition was studied under infinite-dose conditions; the results following the application of the different formulations for 48 h are summarized in Table 6 and Figure 1.

The skin deposition of CBD from the basic PG formulation (see Section 2.3.3) was abnormally high and variable (575.87 ± 357.59 µg/cm^2^). It was also noted that the PG-based formulation was very difficult to remove from the skin surface after a 48 h application time. This was probably due to the phenomenon observed at the end of the experiment, where a “crust” appeared to have been created on the skin surface. Excessive rubbing of the PG formulation resulted in the detachment of the epidermis. Thus, the skin deposition from the PG 5% formulation was considered aberrant and was not discussed further.

The remaining formulations yielded skin depositions between 3.78 ± 0.97 µg/cm^2^ (RP1 1%) and 18.03 ± 5.64 µg/cm^2^ (ACS 2%). The statistical comparison of the different groups is presented in Figure 1. Only the depositions from the two comparator formulations were non-significantly different. Both OS 6.6% and ACS 2% formulations outperformed the reference formulations, RP1 1% and RP2 1%, in terms of cutaneous delivery. The ACS 2% formulation delivered more CBD to the skin than OS 6.6% despite a three-fold-lower CBD content, thus increasing the delivery efficiency.

#### 3.4.3. CBD Transdermal Permeation

CBD permeation across human skin from the control formulations (PG 5%, RP1 1%, and RP2 1%) was low and under the LOD of the analytical method after application for 48 h. The highest CBD permeation was observed for ACS 2% (16.8 ± 5.0 ng/cm^2^) and the flux was 0.30 ± 0.24 ng/cm^2^/h for the linear range of 18-48 h; however, the concentrations were still below the LOQ.

#### 3.4.4. Total CBD Delivery

Total CBD delivery was dominated by cutaneous deposition, given that it was approximately 1000-fold higher than the transdermal permeation (Figure 2).

This skin selectivity limited the amounts of CBD permeated across the skin, and therefore the risk of systemic side effects in vivo. Given that the different formulations have different CBD contents, it is interesting to express the results in terms of delivery efficiency, which can be defined as the proportion of the applied dose that is effectively delivered to the skin (i.e., *Delivery efficiency (%) = 100 × (delivered CBD/applied CBD*)) [88]. The topical delivery efficiency, i.e., the fraction of the applied dose that was deposited in the skin, from the different formulations is presented in Figure 3.

The best delivery efficiency was observed for ACS 2% (0.16 ± 0.04%) followed by RP1 1% and RP2 1%. The delivery efficiency of OS 6.6% was the lowest due to the high CBD content and the fact that the lipophilic nature of the formulation compromised the partitioning of the CBD into the lipophilic intercellular lipid matrix of the stratum corneum. As mentioned above, the percentage of the applied amount of ACS 2% that permeated across the skin was extremely low (1.66 ± 0.62) × 10^−4^%. It is worth noting that infinite-dose experiments underestimate delivery efficiencies in vivo. Since the formulation was applied in excess, the major part of the API content did not come into contact with the skin during the time course of the experiment; thus, delivery efficiencies are more relevant when using more realistic conditions and finite-dosing conditions. This observation was indeed confirmed in the case of CBD during the subsequent finite-dose experiments described below.

### 3.5. CBD Delivery under the Finite-Dose Condition

#### 3.5.1. Experimental Observations

The integrity of all the skin samples was considered to be acceptable since TEWL was below 12.0 g/m^2^h. The measured skin thickness of all the samples ranged between 500 and 750 μm. The gel formulations, CG 1% and CG 2%, were easy to apply to the skin and showed less evaporation and better wetting of the surface compared with the colloidal solution (ACS 2%). Moreover, the skin surface had a good appearance after formulation application for 24 h.

#### 3.5.2. CBD Skin Deposition

Cutaneous delivery experiments under finite-dose conditions (15 mg of formulation applied per cm^2^) were performed following the OECD guidelines [89]. The cutaneous deposition of CBD after the application of the different formulations for 24 h is shown in Figure 4.

Table 7 presents the cutaneous deposition, topical delivery efficiency, and an “improvement factor” (IF) that describes the cutaneous deposition in terms of that observed for RP1, which is given a nominal value of 100. Hence, RP2, which has a 20% greater delivery than RP1, has an IF of 120.

The skin deposition of CBD from the ACS 2%, CG 1%, and CG 2% formulations outperformed RP1 and RP2. There was no statistically significant difference in CBD deposition outcomes between CG 1% and CG 2% (7.87 ± 3.09 and 9.80 ± 1.98 µg/cm^2^, respectively), despite the difference in CBD content. There was therefore no benefit to doubling the concentration of CBD in the formulation from 1% up to 2% from an end-product-manufacturing point of view. It is possible that a larger increase in drug content can nevertheless elicit an increase in the cutaneous deposition if required to improve the therapeutic efficacy. Furthermore, and importantly, there was no difference in the deposition values between CG 1%, CG 2%, and ACS 2%. This indicated that the thermodynamic activity of CBD and its partitioning into the skin were not impaired by gelation and that the CBD was still released successfully.

CG 1% showed the highest topical delivery efficiency, 5.25%, and was statistically superior to RP1 1% and RP2 1% (Figure 5); the 19.4-fold increase observed for the delivery efficiency for ACS 2% from only 0.16% under infinite-dose conditions to 3.11% illustrated that the calculation of this parameter was more relevant under finite-dose conditions, which were also closer to the actual conditions of use in vivo.

#### 3.5.3. CBD Cutaneous Biodistribution

The cutaneous biodistribution method is a technique that enables the drug to be precisely quantified and localized as a function of skin depth, providing more accurate information about drug distribution in the different anatomical regions of the skin [90,91,92,93,94]. Compared to the total deposition analysis, it allowed us to distinguish between the amounts present in the stratum corneum, the outermost barrier of the skin, the epidermis and subjacent structures, such as the dermis, or hypodermis. The amounts present in the region corresponding to the putative site of action could be converted to a concentration to provide a preliminary indication of whether a therapeutic effect was possible. The biodistribution profile of CBD in human skin after a 24 h application of the different colloidal formulations and the reference products were compared in terms of the drug content in each skin lamellae (10 × 40 µm). As presented in Figure 6, the initial observations indicate that the concentration of CBD decreases considerably in the first 200 µm skin depth for all formulations, suggesting that CBD is mostly deposited and localized in the epidermis.

Figure 7 presents the localization of the amounts of CBD in the skin as a function of the different histological skin layers. From this representation, it is possible to identify the superiority of ACS 2% in terms of total topical delivery. This resulted from a higher drug content in the stratum corneum and epidermis with 6.368 ± 3.120 μg/cm^2^ and 5.648 ± 2.997 μg/cm^2^, respectively. The total deposition obtained was 12.953 ± 4.671 μg/cm^2^.

#### 3.5.4. CBD Transdermal Permeation

CBD permeation across human skin was below the LOD of the analytical method after the formulation application for 24 h under finite-dose conditions. However, an underestimation of the CBD uptake by the systemic circulation in vivo could not be entirely excluded. The capillary network is located within the papillary dermis, which is present at a depth of around 150 to 200 µm. However, given the amounts present, the expected plasma concentrations following a single administration should nevertheless remain low.

### 3.6. Comparison with the Existing Data on CBD Transdermal Delivery

The different formulations tested under the conditions used in this study resulted in local delivery of CBD to the skin with cutaneous deposition being far greater than transdermal permeation. The levels of CBD permeated across the skin indicated the low probability of significant transdermal permeation detectable in vitro when using realistic formulation application conditions, e.g., such as those used for the finite-dose experiments. The much smaller amounts of formulation applied also eliminated the effect of occlusion due to the amount of formulation applied to the skin surface under infinite-dosing conditions.

The methodologies used for previous in vitro investigations of the transdermal delivery of CBD are heterogeneous and difficult to compare [95,96,97,98]. The heterogeneity comes first from the skin type and thickness and second from the experimental conditions, in particular, the composition of the receiver phase. Studies using thin skin and aggressive receiver solutions show the highest transdermal CBD permeation; however, the predictive power of those studies for behavior in vivo are open to discussion (see below).

Franzè et al. used similar (but not identical) conditions to those used in the present study (e.g., formulation application for 24 h and skin samples with a thickness of 740 µm, but porcine skin was used instead of human skin) [96] and the skin deposition of CBD observed with ACS 2%, CG 1%, and CG 2% (~8–10 μg/cm^2^) was similar to the amounts of CBD in the dermis following the application of the “Drug-in-Micelles-in-Liposomes” (DiMiL) formulations that they developed (~8–13 μg/cm^2^). This was achieved in the present study, despite the use of finite-dose conditions with smaller volumes and lower amounts of CBD applied per unit surface area (~15 μL/cm^2^ with a CBD content of 150 μg/cm^2^ for CG 1% and 300 μg/cm^2^ for ACS 2% and CG 2%, versus 472 μL/cm^2^ with a CBD content of 358 μg/cm^2^ for the DiMiL formulations). The only question that arose with respect to that study, in our opinion, was the amount of Kolliphor HS 15 (2.5% *w*/*v*) present in the receiver phase; Kolliphor HS 15 was also used in the DiMiL formulation, so it was probable that the skin was impregnated with Kolliphor and this could have enhanced CBD transport (as evidenced by the transdermal permeation). Stinchcomb et al. performed flow-through studies using saturated CBD solutions (in mineral oil, or propylene glycol:water (7:3) and propylene glycol:water:ethanol (4:5:4)) and thin cadaver skin, but lower contents of solubilizer (Brij 98 0.5% *w*/*v*) in the receiver [97]. The transdermal flux observed in those studies could be explained by the thermodynamic activity of the CBD in the basic formulations, the skin thickness, and the fact that its barrier function could have been compromised due to the contact with propylene glycol and/or ethanol for 48 h. It is worth noting that the inclusion of ethanol increased CBD transdermal flux.

Two recent reviews have summarized the developments in CBD dermal and transdermal delivery [99,100]. Lodzki et al. administered CBD to nude mice from an ethosomal formulation applied for 24 h. The results showed high blood levels of between 600 and 1000 ng/mL without any flux calculations [48]. However, nude mice skin is much thinner than human skin and is considered to be a very permeable surrogate. Paudel et al. studied the pharmacokinetics of CBD when administered nasally and transdermally to guinea pigs [101]. The CBD (1.8%) gel formulation for transdermal delivery was made from an 80:20 PG:H_2_O solution with or without 6% Transcutol P, gellified by hydroxyethyl cellulose. For transdermal studies, 500 μL of the CBD gel formulation with or without the enhancer was applied to the dorsal region. Patches were constructed to keep the formulation in place on an area of 13.2 cm^2^ for 48 h. The flux of CBD without Transcutol P was 6.13 ± 0.43 nmol/cm^2^/h (1927.7 ng/cm^2^/h). Interestingly, CBD levels remained at a steady state after the patch removal, indicating a possible reservoir effect of the skin.

### 3.7. Clinical Relevance

The majority of in vitro and in vivo preclinical studies investigating the use of the skin as a delivery route for the administration of CBD focus on its transdermal delivery for systemic applications. The in vitro studies used infinite-dose conditions and, given the physicochemical properties of CBD, aggressive solubilizers were required to maintain sink conditions in the receiver phase. Solubilizers can affect the fragile skin membranes used in the experiments, e.g., heat-separated epidermis or very thin dermatomed skin (thickness <300 µm), increasing permeability and resulting in increased amounts of CBD being detected in the receiver compartment, leading to a risk of overestimating systemic delivery in vivo. The present study, where a more “realistic” receiver phase composition was used with a thicker membrane (500–700 µm), resulted in a much more reduced transdermal delivery of CBD (ACS 2%: cumulative permeation—16.8 ± 5.0 ng/cm^2^) than that observed in other studies. The use of a thicker skin membrane in the present study as compared to other reports could result in an underestimation of transdermal permeation, since entry into blood capillaries is possible in the upper dermis. On the other hand, the use of a less aggressive receiver phase in the present study might decrease the risk of overestimating systemic levels achieved after delivery across human skin in vivo.

A recent clinical trial assessing the transdermal delivery of CBD (and THC) using an emulsion-based formulation after application of a 100 mg CBD dose to the hand, wrist, and forearm, and where blood levels were monitored for 12 h, showed that the highest CBD concentration was ~0.5 ng/mL [102]. For comparison, the study by Knaub et al., which used a self-emulsifying drug delivery system for the oral delivery of 25 mg of CBD, achieved maximum concentrations in the range of 6-20 ng/mL [79]; thus, despite the 4-fold lower dose, the oral administration resulted in 12–40-fold-greater levels in the blood. Given the plasma concentrations observed in humans after oral (and transdermal) administration, we decided for the present study to focus on topical delivery with a view to local administration for the targeted treatment of dermatological indications. Since, to the best of our knowledge, there were no data about finite-dose CBD applications, a better predictor of in vivo behavior, it was decided to introduce this into the study plan and to conduct finite-dose-application experiments (without occlusion) together with an investigation of the biodistribution profile to determine which regions of the epidermis and dermis could be targeted.

It has been suggested that the topical application of CBD can decrease levels of proinflammatory cytokines, e.g., IL-6 and IL-17, and promote the release of anti-inflammatory cytokines, e.g., IL-10 [103]. From this perspective, it is reasonable to assume that topical applications can be used for CBD delivery to the viable epidermis, thereby targeting atopic dermatitis [4,13,14,15,16], pruritus [7,12,25,26,27,28,29,30,31,32], and perhaps psoriasis [7,33] and other inflammation-related diseases.

In conclusion, the results of this study demonstrate the feasibility of delivering CBD to the epidermis and upper dermis using a patient-friendly formulation under finite-dose conditions. The data presented here were collected using healthy human skin, and it is clear that a dermatological condition, such as atopic dermatitis, could impair skin barrier function and thereby increase cutaneous bioavailability. Thus, the effect of barrier impairment due to disease on CBD delivery and biodistribution needs to be investigated, and this is envisaged in subsequent studies.

## Figures and Tables

**Figure 1 pharmaceutics-16-00202-f001:**
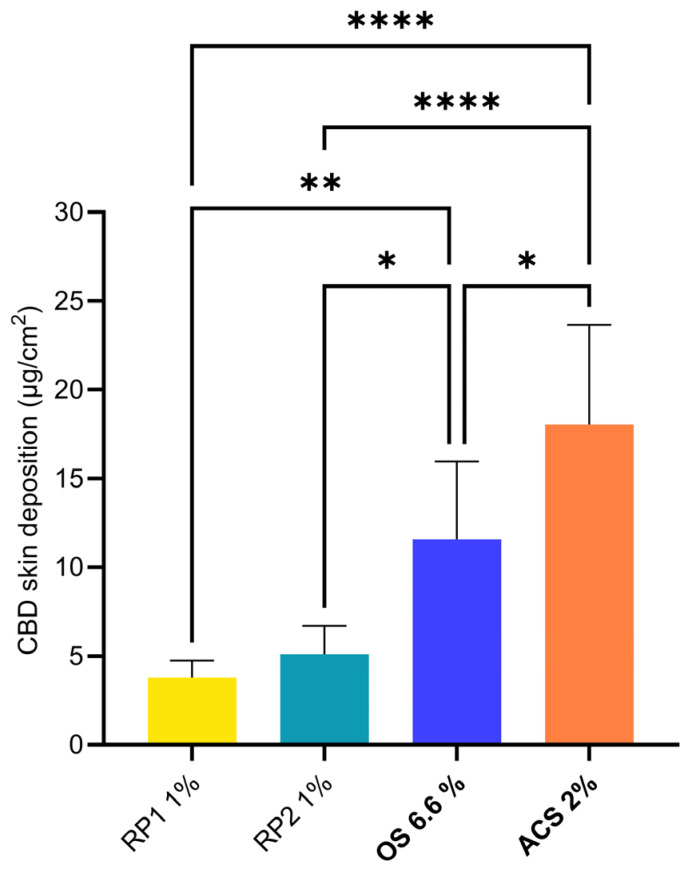
Skin deposition of CBD from different formulations (defined in Table 3) under infinite-dose conditions. Statistical comparison of CBD skin deposition from the different formulations. *p*-values were calculated using the ANOVA one-way test; statistically significant differences are denoted by asterisks (* *p* ≤ 0.05; ** *p* ≤ 0.01; **** *p* ≤ 0.0001).

**Figure 2 pharmaceutics-16-00202-f002:**
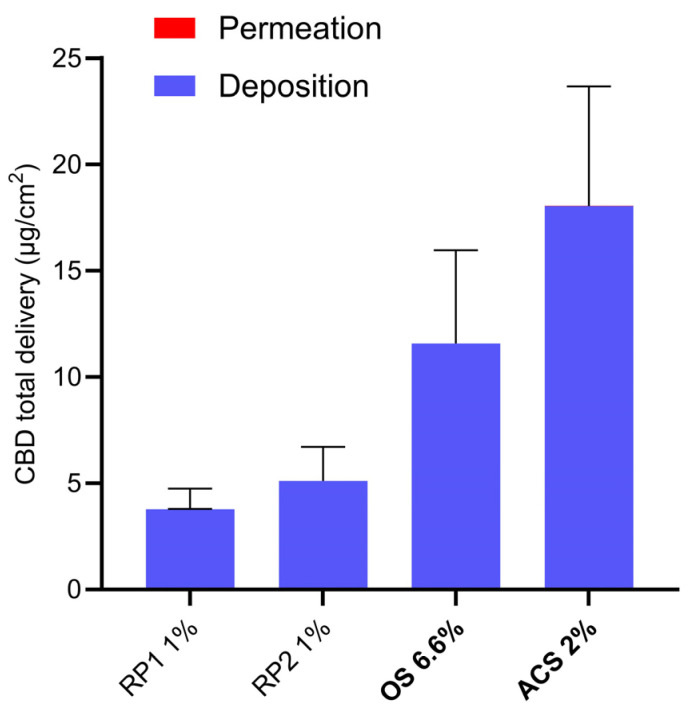
CBD total delivery from different formulations (defined in Table 3) under infinite-dose conditions. Given the 1000-fold selectivity for skin deposition (blue), this dominates the total delivery, and it is only possible to see a thin red line (indicating permeation) for ACS 2%.

**Figure 3 pharmaceutics-16-00202-f003:**
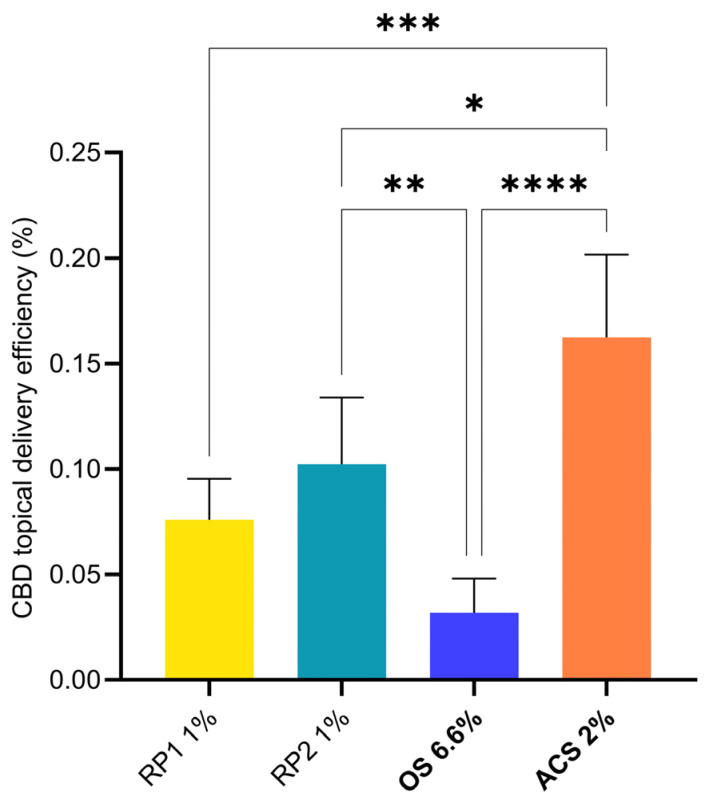
Statistical comparison of topical delivery efficiencies (defined in Table 3) under infinite-dose conditions. *p*-values were calculated using the ANOVA one-way test; statistically significant differences are denoted by asterisks (* *p* ≤ 0.05; ** *p* ≤ 0.01; *** *p* ≤ 0.001; **** *p* ≤ 0.0001).

**Figure 4 pharmaceutics-16-00202-f004:**
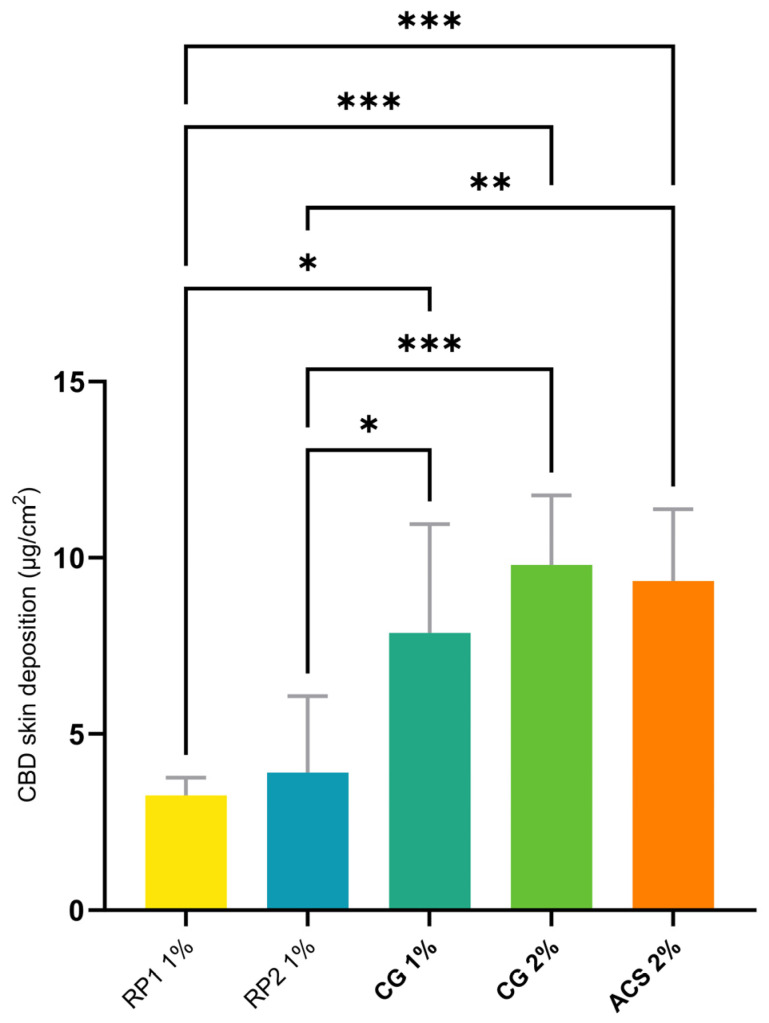
Statistical comparison of skin deposition of CBD from different formulations (defined in Table 4) under finite-dose conditions. *p*-values were calculated using the ANOVA one-way test; statistically significant differences are denoted by asterisks (* *p* ≤ 0.05; ** *p* ≤ 0.01; *** *p* ≤ 0.001).

**Figure 5 pharmaceutics-16-00202-f005:**
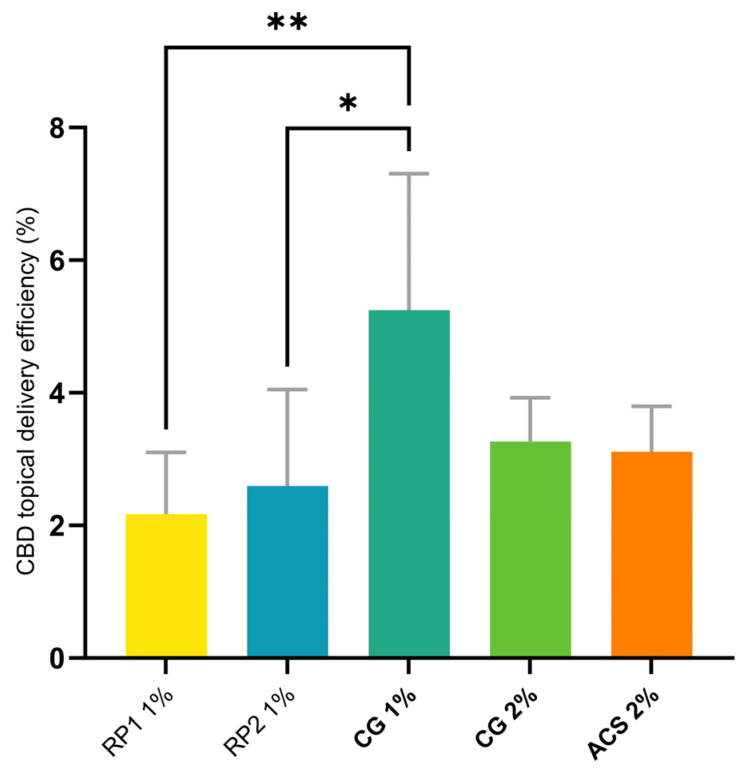
Statistical comparison of total topical delivery efficiencies of CBD from different formulations (defined in Table 4) in finite-dose conditions. Topical delivery efficiency represents the proportion of the applied dose that is delivered to the skin. *p*-values were calculated using the ANOVA one-way test; statistically significant differences are denoted by asterisks (* *p* ≤ 0.05; ** *p* ≤ 0.01).

**Figure 6 pharmaceutics-16-00202-f006:**
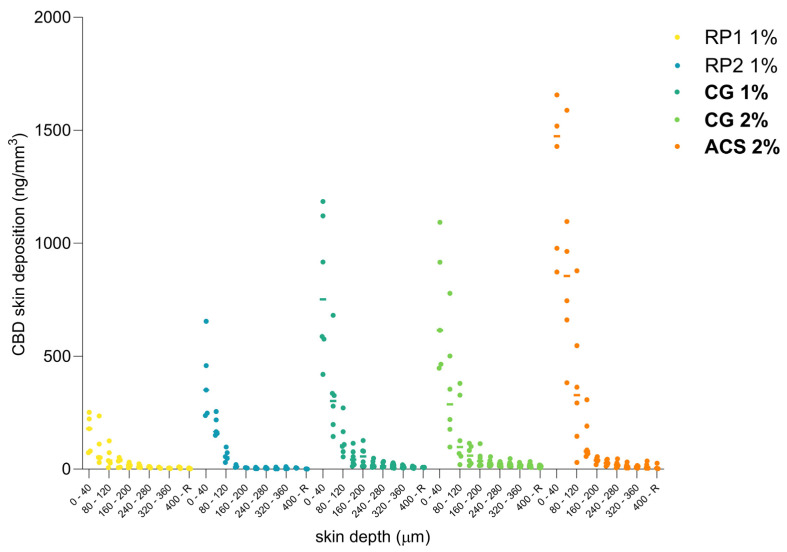
CBD cutaneous biodistribution profiles of CBD in human skin lamellae (10 × 40 µm) from the colloidal formulations and the reference products (defined in Table 4) under finite-dose conditions.

**Figure 7 pharmaceutics-16-00202-f007:**
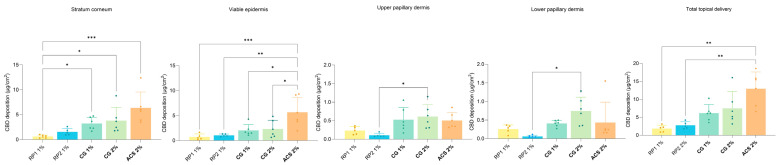
Cumulative CBD deposition by anatomical skin layer under finite-dose conditions. *p*-values were calculated using the ANOVA one-way test; statistically significant differences are denoted by asterisks (* *p* ≤ 0.05; ** *p* ≤ 0.01; *** *p* ≤ 0.001).

**Table 1 pharmaceutics-16-00202-t001:** Gradient elution of CBD and its internal standard CBD-d3.

Time (min)	Flow Rate (mL/min)	%A	%B
0	0.3	30	70
0.5	0.3	30	70
3.5	0.3	0	100
4.0	0.3	0	100
4.1	0.3	30	70

**Table 2 pharmaceutics-16-00202-t002:** MS/MS settings for the detection of CBD and its internal standard CBD-d3.

	Cannabidiol		Cannabidiol-d3	
Nature of parent ion	[M + H]^+^		[M + H]^+^	
Mass transitions	1	2	1	2
Precursor ion (*m*/*z*)	315.23	315.23	318.30	318.30
Product ion (*m*/*z*)	123.00	193.14	123.06	196.20
Dwell time (s)	0.249	0.249	0.249	0.249
Cone voltage (V)	12.0	12.0	12.0	12.0
Collision energy (V)	28.0	14.0	32.0	18.0
Capillary voltage (kV) (ESI positive)	2.0			
Desolvation temperature (°C)	350			
Desolvation gas flow (L/h)	650			
Collision gas flow (L/h)	0.15			
LM resolution 1	15.0			
HM resolution 1	15.0			
LM resolution 2	15.0			
HM resolution 2	15.0			

**Table 3 pharmaceutics-16-00202-t003:** Details of the groups and experimental conditions for the infinite-dose studies.

Group						
	A	B	C	D	E	F
Description	5% PG ^a^	RP1 ^b^	RP2 ^c^	**OS ^d^ 6.6%**	**ACS ^e^ 2%**	Blank
Formulation	PG	Cream	Serum/face oil	Oil	Colloidal solution	PBS
Volume/mass applied (g)	1					
CBD concentration	5%	1%	1%	6.6%	2%	0
Receiver pH	7.4					
Applied CBD amount (mg/cm^2^)	25	5	5	33	10	0
N	6	6	6	6	6	1
Type of skin	Sliced human skin: 500–700 μm					
Contact area (cm^2^)	2					
Receiver phase	PBS + 0.5% Tween 80					
Sampling (h)	10, 12, 18, 24, 36, 42, 48					
Sampling volume (μL)	300					
Extraction conditions	10 mL of MeOH:water 4 h					

^a^ 5% in propylene glycol; ^b^ Charlotte’s Web Cream 1% (RP1); ^c^ Lord Jones Serum (face oil) 1% (RP2); ^d^ 6.6% oil solution (OS); ^e^ 2% Aqueous Colloidal Solution (ACS).

**Table 4 pharmaceutics-16-00202-t004:** Details of the groups and experimental conditions for the finite-dose studies.

	Group					
	A	B	C	D	E	F
Description	RP1 ^a^	RP2 ^b^	**ACS ^c^ 2%**	**CG ^d^ 1%**	**CG ^e^ 2%**	Blank
Formulation	Cream	Serum/face oil	Colloidal solution	Colloidal gel	Colloidal gel	PBS
Volume/mass applied (mg)	30					
CBD concentration	1%	1%	2%	1%	2%	0
Receiver pH	7.4					
Applied CBD amount (μg/cm^2^)	150	150	300	150	300	0
N	6	6	6	6	6	1
Type of skin	Sliced human skin: 500–700 μm					
Contact area (cm^2^)	2					
Receiver phase	PBS + 0.5% Tweens 80					
Sampling (h)	10, 16, 20, 24					
Sampling volume (μL)	300					
Extraction conditions	10 mL of MeOH:water 4 h					
Cutaneous biodistribution	10 lamellae measuring 40 μm down to a depth of ~400 μm					

^a^ Charlotte’s Web Cream 1% (RP1); ^b^ Lord Jones Serum (face oil) 1% (RP2); ^c^ 2% Aqueous Colloidal Solution (ACS); ^d^ 1% Colloidal Gel (CG); ^e^ 2% colloidal gel.

**Table 5 pharmaceutics-16-00202-t005:** Characterizations of the ACS 2% and CG 2% CBD formulations.

Time Point (Months)	ACS 2%			CG 2%		
	z Average (nm)	PDI	CBD (%)	z Average (nm)	PDI	CBD (%)
0	45.70	0.071	2.100 ± 0.004	47.08	0.129	2.064 ± 0.004
12	47.83	0.070	2.127 ± 0.004	52.70	0.175	2.104 ± 0.021
24	47.99	0.072	2.121 ± 0.003			

**Table 6 pharmaceutics-16-00202-t006:** Cutaneous deposition of CBD from different formulations under infinite-dose conditions.

Formulation	Cutaneous Deposition (μg/cm^2^)
	Mean ± SD
Blank	NA
RP1 ^a^ 1%	3.78 ± 0.97
RP2 1%	5.12 ± 1.59
**OS 6.6%**	11.57 ± 4.40
**ACS 2%**	18.03 ± 5.64

^a^ See Table 3 for formulation codes.

**Table 7 pharmaceutics-16-00202-t007:** Cutaneous deposition, topical delivery efficiency, and improvement factor upon CBD delivery from different formulations under finite-dose conditions.

Formulation	Cutaneous Deposition (μg/cm^2^)	Topical Delivery Efficiency	Improvement Factor ^a^
	Mean ± SD	%	%
Blank	NA	NA	NA
RP1	3.25 ± 0.51	2.17. ± 0.94	100 (ref)
RP2	3.90 ± 2.18	2.60 ± 1.45	120
**CG 1%**	7.87 ± 3.09	5.25 ± 2.06	242
**CG 2%**	9.80 ± 1.98	3.26 ± 0.66	302
**ACS 2%**	9.34 ± 2.04	3.11 ± 0.68	287

^a^ Defined as the increase in cutaneous deposition as compared to RP1.

## Data Availability

The data presented in this study are available on request from the corresponding author.

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
