# Peer review of "Cutaneous Delivery and Biodistribution of Cannabidiol in Human Skin after Topical Application of Colloidal Formulations"

_pharmaceutics, 2024, doi:10.3390/pharmaceutics16020202_

Round 1
Reviewer 1 Report
Comments and Suggestions for Authors
The idea is interesting and it has a scientific appeal; however, the study seems a little preliminary. Few experiments and data were shown to support any conclusions or future applications of the formulation. Other major comments:
- - In the UHPLC-UV method: Shouldn't the retention time be higher? Considering both the lipophilicity of CBD and the column dead volume, which can impair the analysis
- - Have you not performed encapsulation efficacy of ACS formulation? How was the drug content calculated in item 3.2.1? Do you consider that all the CBD was encapsulated in the nanoparticles?
- - Did you perform more characterization tests with ACS formulation? Like stability or in vitro release. It is really important to ensure the quality of the formulation before proceeding with skin permeation studies.
- - Overall the results presentation are a little confusing. There are a lot of numbers through the text, making the reading more difficult.
- - Especially regarding to the formulations: results were poorly presented (missing tables or graphs related to the characterization)
- - The formulation of oil solution group was not clear
Comments on the Quality of English LanguageEnglish is fine
Reviewer 2 Report
Comments and Suggestions for Authors
The paper describes the development and in vitro test of topical formulations for the delivery of CBD to the skin. The manuscript is very well written, clearly organized and the data are deeply discussed and compared with literature data.
There are two points that need to be taken into account:
1. Lines 161-169 colloidal gel formulation: did the author check the effect of gelling (agent) on droplet size and distribution?
2. Lines 338-344: the calculation of delivery efficiency in experiments performed in infinite dose conditions is questionable, due to the very small amount recovered in comparison with the amount applied. Infact the amount of CBD recovered is in the order of 5-20 µg/cm2, whereas the amount of CBD applied varies from 5 mg/cm2 (1% formulation) to 33 mg/cm2 (6.6% formulation).
Reviewer 3 Report
Comments and Suggestions for Authors
In this research, the biovailability of topical CBD formulations is probed.
The theme of the work is relevant with significant contribution to scientific research. The methodologies used were adequate for the proposed study. The results obtained generally prove the proposed objectives, and the conclusions are corroborated by the data presented.
However, in my opinion, there are a few points that should be further clarified:
- Permeation studies were conducted using PBS and 0.5% Tween 80. Have any studies on the perturbation on skin barrier caused by this concentration of surfactant been performed?
- Why was the blank performed on just 1 replicate? Where are these results presented?
- At the end of the experiment, a small area of 0.5 cm2 was punched out from the 2 cm2 skin samples. Why was this approach followed?
- Where can the stability studies results of ACS 2% mentioned in section 3.2.1 be found (24 months)?
- Table 5 and Figure 1 are showing the same data. In my opinion, presentation in table format is redundant.
- Authors should comment the very high SD in section 3.4.3, since it is almost as high as the mean flux values.
- The results mentioned in section 3.4.4 are not found in table 6, and only in Figure 3, unlike what is stated at the end of the paragraph.
